# Conditions for Production of Composite Material Based on Aluminum and Carbon Nanofibers and Its Physic-Mechanical Properties

**DOI:** 10.3390/nano9040550

**Published:** 2019-04-04

**Authors:** Oleg V. Tolochko, Tatiana S. Koltsova, Elizaveta V. Bobrynina, Andrei I. Rudskoy, Elena G. Zemtsova, Sergey O. Kirichenko, Vladimir M. Smirnov

**Affiliations:** 1Peter the Great St. Petersburg Polytechnic University, Polytechnicheskaya 29, 195251 St. Petersburg, Russia; annelet@yandex.ru (T.S.K.); bobrynina@inbox.ru (E.V.B.); rector@spbstu.ru (A.I.R.); 2Saint Petersburg State University, Universitetskii pr.26, 198504 St. Petersburg, Russia; ezimtsova@yandex.ru (E.G.Z.); sergey.kirichenko@spbu.ru (S.O.K.); vms11@yandex.ru (V.M.S.)

**Keywords:** Al-CNF composite, in situ synthesis of CNF, mechanic activation, carbon nanofibers, durability, plasticity, thermal conductivity

## Abstract

Aluminum-based metallic matrix composites reinforced by carbon nanofibers (CNFs) are important precursors for development of new light and ultralight materials with enhanced properties and high specific characteristics. In the present work, powder metallurgy technique was applied for production of composites based on reinforcement of aluminum matrices by CNFs of different concentrations (0~2.5 wt%). CNFs were produced by chemical vapor deposition (CVD) and mechanical activation. We determined that in situ synthesis of carbon nanostructures with subsequent mechanic activation provides satisfactory distribution of nanofibers and homogeneous composite microstructure. Introduction of 1 vol% of flux (0.25 NaCl + 0.25 KCl + 0.5 CaF_2_) during mechanic activation helps to reduce the strength of the contacts between the particles. Additionally, better reinforcement of alumina particles and strengthening the bond between CNFs and aluminum are observed due to alumina film removal. Introduction of pure aluminum into mechanically alloyed powder provides the possibility to control composite durability, plasticity and thermal conductivity.

## 1. Introduction

Metal matrix composites reinforced with carbon nanotubes (CNTs) and carbon nanofibers (CNFs) have become the subject of many studies [1]. This is due to the unique properties of CNTs [2,3]. Uniform distribution of carbon nanotubes in the metal matrix remains a challenge due to their high propensity for agglomeration. The main technological problems in the synthesis of hybrid composite materials containing CNTs or CNFs are the distribution of the hardening phase in the volume of the composite, the strength of its bonding with the matrix, and the chemical and structural stability of the carbon ordered structures in the composition. These problems are solved by researchers using various methods and, in the first turn, in the stage of preparation of the composite powder [4,5,6,7,8,9,10,11,12,13,14,15,16,17,18,19,20,21,22,23]. Most traditional methods of mixing of the matrix and CNTs powders are a mechanical grinding in a ball mill [4,5,6,7,8], flake powder metallurgy [9,10] molecular level mixing [11], in situ CNT growth on metal powders [12,13,14], and spray drying of small metal particles with CNTs [15]. To create metal-based compact materials reinforced by CNTs, technologies of powder metallurgy [16,17], liquid metallurgy [18], galvanic plating [19], sintering in spark plasma [20,21], and thermal spraying are used [22,23].

A significant problem here is the difficulty to reach a balance between strength and ductility for Al-CNT composites. If the enhanced tensile strength was obtained by addition of CNTs, the ductility of composites was low [24,25,26]. There is also a risk of low electrical and thermal properties due to the interaction of carbon nanostructures with the matrix.

In the current paper, we investigate the possibility to produce the aluminum-based composites reinforced with carbon nanostructures with different CNF content (0~2.5 wt.%) by powder metallurgy and mechanical activation. Carbon nanostructures were obtained by chemical vapor deposition (CVD) directly onto matrix aluminum particles (in situ) with subsequent milling of composite in planetary ball mill. We expected that such approach leads to uniform distribution of dispersed phase CNFs in aluminum matrix along with strong bond between CNF and aluminum matrix phases.

## 2. Materials and Methods

### 2.1. Materials

The initial material used was pulverized aluminum powder of PA-4 mark, GOST 6058-73 standard with particle size less than 120 µm and purity of 99.5 wt.%. Amounts of the main impurities (Si, Fe, Cu) do not exceed 0.4, 0.35, and 0.02 wt.%, respectively.

The powder particles had a teardrop shape (a) and a rough surface (b) (Figure 1).

### 2.2. Preparation of Aluminum-CNFs Composites

To grow carbon nanostructures, a nickel- or cobalt-containing catalyst with 0.035 wt.% Ni or Co was deposited on the powder surface. The metal source was either Ni(NO_3_)_2_·6H_2_O or Co(NO_3_)_2_·6H_2_O that are highly soluble in water and decompose to NiO and CoO at 300–350 °C. The aluminum powder was mixed with a 0.01% aqueous solution of Ni or Co nitrate (10 mL per 1 g powder). The solution was stirred for 10 min and then carefully dried at 100 °C in a drying chamber. Then, the powder was heated additionally in a hydrogen atmosphere for complete reduction of salts to metallic nickel or cobalt. The reactor was blown with argon before and after the synthesis. H_2_/C_2_H_2_ ratio in the gas mixture during the synthesis was 8.31. The temperature of the synthesis was 550 °C. The carbon content was determined by weighing the sample before and after synthesis. The length of carbon nanostructures was determined visually by SEM images.

Mechanical activation of the powder was carried out in a micro-mill PULVERISETTE 7 premium line. The 2.5 wt.%-Al-CNFs composite powders (with the addition of 1 vol.% flux 0.25 NaCl + 0.25 KCl + 0.5 CaF_2_) were placed in 80 mL stainless steel jars containing stainless steel balls of 10 mm diameter (ball-to-powder ratio = 10:1). The powders were milled under argon at 500 rpm for 180 min. For variation of CNFs content, the mechanically activated Al-CNFs samples were mixed with a certain amount of pure Al powder at 500 rpm for 180 min.

### 2.3. Powder Compaction

Materials with a carbon content of 0.5 to 2.5 were used for pressing. The specimens were compacted by cold pressing at a pressure of 400 MPa with subsequent heating with mold up to 480 °C with final hot pressing at 600 MPa. Three samples were made for each measurement.

### 2.4. Characterization

The structure and morphology of the powder materials was studied by scanning electron microscopy (TESCAN Mira-3M, Brno-Kohoutovice, Czech Republic).

Metallographic studies were carried out using an optical microscope (Carl Zeiss Observer D1m, Feldbach, Switzerland). The thermal conductivity of the specimens was computed from the measured values of the thermal diffusivity by equation:λ = αρc_p_(1)
where c_p_ ([c_p_] = J(g·K)) is the specific heat capacity of the specimen. The accuracy of the measurements (datasheets of manufacturer) was 2.3% for the thermal diffusivity, 4% for the heat capacity, and 5% for the thermal conductivity.

The density of the sintered specimens was determined by hydrostatic weighing.

### 2.5. Mechanical Tests

Brinell hardness was measured by hardness tester Zwick/Roell ZHU (Ulm, Germany) using a 5 mm steel ball at a load of 98 N. Bending tests were carried out in accordance with GOST 14019-80 “Metals and alloys. Methods of testing for bending”. Vickers microhardness was measured with a Buehler Micromet Hardness Tester (Dusseldorf, Germany) data obtained using a 1 g load, with a 10 s dwell time.

### 2.6. Specific Surface Area and Porosity Measurement

Specific surface area was calculated from physical adsorption data that were obtained using volumetric analyzer-porometer Micromeritics ASAP2020 MP (Micromeritics Instrument Corporation, Norcross, GA, USA). Preliminary adsorption isotherms measurements were carried out using standard nitrogen adsorption technique at 77 K; specific surface areas were calculated by BET and total pore volumes, available for adsorption (so-called Gurvich volume) at relative pressure of 0.995. Due to the small specific surface area of the samples, additional measurements using krypton as an adsorbate were carried out at 77 K to clarify the values of specific surfaces; the specific surface was calculated using BET. Before tests, the specimens were vacuum degassed at 250 °C for at least 14 h.

### 2.7. Raman Spectral Studies

Raman spectra were recorded using Raman spectrometer Bruker Senterra T64000 (Bruker Optics Inc., Billerica, MA, USA) with excitation wavelength of 488 nm in the range 100–3500 cm^−1^ Spectra were normalized using the SVN technique with subsequent baseline correction.

## 3. Results and Discussion

To study of the specific surface area and porosity of the initial samples of aluminum with Cups on the surface, three samples of each composition were studied (Table 1). Measurements showed that CNFs assist surface increase. Flux introduction leads to decrease specific surface area. It can be explained by the blocking of surface available for adsorbate by flux particles.

Raman spectra of all three specimens of aluminum with CNF-modified surface (Figure 2, Figure 3 and Figure 4) are characterized by typical peaks for graphite-like materials: G-peak at circa 1580 cm^−1^ corresponding to intraplanar C–C bonds vibrations and D-peak at circa 1350 cm^−1^ indicating defects and disorder in carbon nanostructures. Additionally, a wide 2D(G*) peak at 2500–2800 cm^−1^ (two-phonon process of the second order) is a characteristic of graphite-like materials containing sp^2^-hybridized carbon.

To obtain good distribution of carbon nanostructures in the matrix, we deposited a nickel and cobalt catalyst onto the surface of the aluminum particles from aqueous solutions. Right before the synthesis, the specimens coated with the catalyst were annealed additionally in a hydrogen environment for 10 min at 550 °C to provide decomposition of the nickel or cobalt nitrate and reduction of the oxide to metallic nickel or cobalt that served as catalyst of carbon nanofibers growth on the aluminum powder surface. The content of the metal catalyst was 0.02%.

Carbon nanostructures were synthesized at 550 °C for 5–20 min. Figure 5 presents the dependence of the specimen mass variation with respect to the initial weighed portion of aluminum on the synthesis time.

With increase of the synthesis time, the growth rate of the carbon nanostructures decreases due to catalyst deactivation, i.e., disappearance of the dominant nucleation places of carbon nanostructures on the powder surface. Nickel catalyst provides a greater amount of carbon. Deactivation of cobalt catalyst occurs faster than nickel one. Rapid deactivation of the catalyst resulted in the production of Co nanofibers with a shorter length. Figure 6 shows SEM images of synthesized aluminum-carbon nanofibers composites.

If we compare the dependence of weight gain with SEM images, one can see for 10 min samples that the structures are similar and carbon amount is approximately equal. Increasing the synthesis time to 20 min leads to the fact that the sample with a nickel catalyst has a higher carbon content, and the length of the carbon structures is greater. The co-catalyst differs from the Ni-containing one. The decontamination process in the case of cobalt occurs quicker, and after 10 min, significant weight gain was not observed. When nickel was used, the increase in carbon content was due to an increase in the fibers length (Figure 6c,d). Next, we conducted experiments on powders with nanotubes obtained on nickel and cobalt catalysts. At equal carbon content, physical and mechanical characteristics coincided. Variations of nanotubes length must influence the physic mechanical properties; however, to achieve a big difference in length with equal contents with our technique is quite a difficult task.

At the next stage, a powder with short CNFs was selected for further mechanical activation. Mechanical activation of the powder was carried out with the aim of reinforcing the particles throughout the volume by carbon nanofibers.

Grinding was carried out under argon either without any additives or with addition of flux. Flux was added to destroy the oxide film on the surface of the original aluminum particles and to prevent significant welding of particles.

The KCl-NaCl-CaF_2_ system flux was introduced to break the oxide film on the aluminum surface during the grinding process. We used Al powder with 2.5 ± 0.2 wt.% CNFs content.

The results are shown in Figure 7. Particles up to 500 µm in size that have almost spherical shape were obtained without flux. With the flux particle size is less than 200 μm and has a plate-like shape.

After treatment in a planetary mill, the particles had a coarse morphology and a size of 50–200 μm (Figure 7a,b). Investigation of the microstructure of the composite particles shows a good carbon distribution (Figure 8a,b). The microhardness of particles was 200 HV (microhardness of the particles mechanically activated without flux was 100–120 HV). A significant increase in hardness when using flux can be explained by the formation of stronger bond between the CNFs and aluminum due to oxide film removal.

To vary the carbon content in this study, pure aluminum was added to the powder after the mechanical activation and mixed for additional 15 min under the same conditions. Thus, samples with 1 and 0.5 wt.% carbon were obtained. The relative density of the samples, determined by the method of hydrostatic weighing, was not less than 97%. Hardness and thermal conductivity of materials were investigated (Figure 9). For comparison, we provide data for the specimens compressed immediately after synthesis and ones not subjected to mechanical activation [27].

Hardness of the material with 2.5 wt.% CNF reached 180 HV. Its thermal conductivity was 60 W/(m·K), that corresponds approximately to the value of the specimens with 1–2 wt.% CNF without mechanic activation. These low values are explained by the appearance of a thermal barrier at the interface and by the aluminum carbide formation [27]. Dilution of the composite powder with pure aluminum leads to a hardness decrease and a sharp thermal conductivity increase. At 0.5–1% CNF, thermal conductivity of 160–180 W/(m·K) is circa 70% from the value for pure Al (237 W(m·K) at 300 K [28]). If nanofibers are located at the surface, thermal conductivity was 96 W(m·K) at similar hardness (circa 60 HV). The increase in thermal conductivity is due to increased Al–Al contact in the composite.

One of the promising areas of application for aluminum composites based on low thermal conductivity is its use as a structural material for the manufacture of enclosures for electronic devices and lithium-ion batteries; in this case, devices operate at temperatures below 0.

To determine the plastic characteristics, the composites were tested for bending. Figure 10a–d illustrates the bending test results along with composites structures.

The composite with a uniform fine structure (Figure 10b) has a sufficiently high bending strength (485 MPa); at the same time, plasticity is practically absent. Composites diluted with pure Al (Figure 10c,d) look like grains of pure aluminum surrounded by reinforced particles. At small CNFs amount, matrix particles are deformed and represent plates. The strength is reduced significantly down to 310 and 165 MPa for 1 and 0.5 wt.%, respectively. Corresponding relative elongation was 4 and 5.8%. Low strength at a relative elongation of 5.8% (sample with 0.5 wt.%) suggests that additional compacting is required by hot rolling or extrusion.

Thus, during the study we determined that the in situ synthesis of nanostructures and subsequent mechanical activation provides a good distribution of nanofibers and a homogeneous microstructure of the composite.

The introduction of a flux during mechanical activation contributes to the reduction of adhesion between the particles, the better reinforcement aluminum particles and the connection between the CNFs and aluminum due to the oxide film removal.

The introduction of pure aluminum particles into the mechanically alloyed powder allows varying the strength, ductility and thermal conductivity of the composite for specific applications. This approach will provide a uniform distribution of the dispersed phase in the aluminum matrix and strengthen the binding between CNF and the matrix.

## 4. Conclusions

The paper presents the study of the powder metallurgy production of aluminum-based composites, reinforced by carbon nanostructures with different amount of CNFs. The main conclusions are as follows:(1)Gas phase technique for carbon nanostructures catalytic synthesis directly on the Al microparticles surfaces allows to gain uniform distribution of carbon in the matrix. Deactivation of cobalt catalyst starts earlier than that of nickel catalyst; however, with an equal carbon content, the type of catalyst does not affect the physical and mechanical characteristics.(2)Mechanic activation provides good nanofiber distribution as well as homogeneous composite microstructure. The introduction of flux during mechanical activation helps to reduce the weldability of particles; additionally, better reinforcement of aluminum particles and the connection between CNFs and aluminum are reached by removing the oxide film.(3)Strength, ductility, and thermal conductivity of the composite can be varied by introducing CNF to pure aluminum in different concentrations.

## Figures and Tables

**Figure 1 nanomaterials-09-00550-f001:**
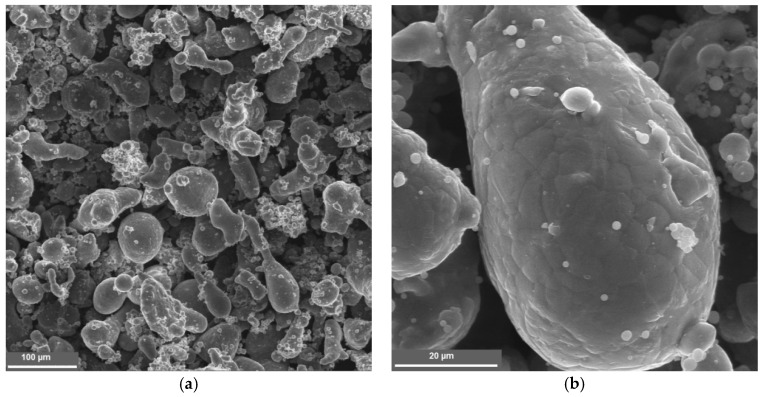
Microstructure of the composite particle: (**a**) a teardrop shape and (**b**) a rough surface.

**Figure 2 nanomaterials-09-00550-f002:**
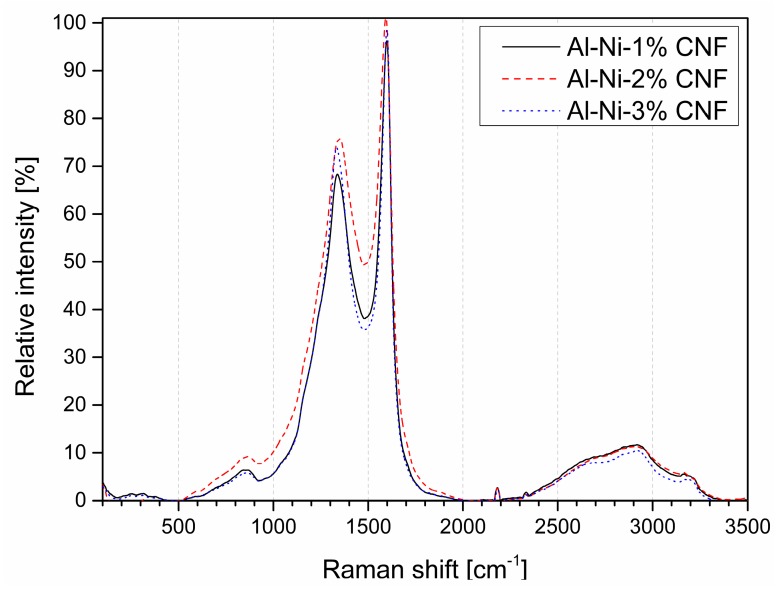
Raman spectra of aluminum with various weight content of CNF.

**Figure 3 nanomaterials-09-00550-f003:**
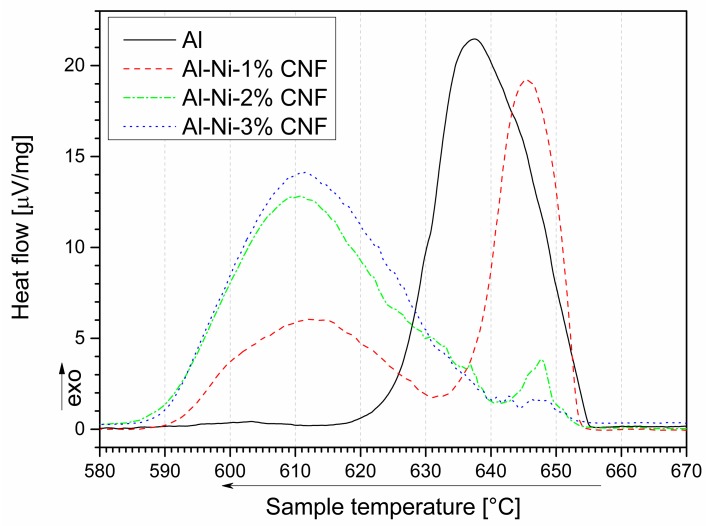
DSC curves of aluminum with various CNF content during cooling near Al melting point.

**Figure 4 nanomaterials-09-00550-f004:**
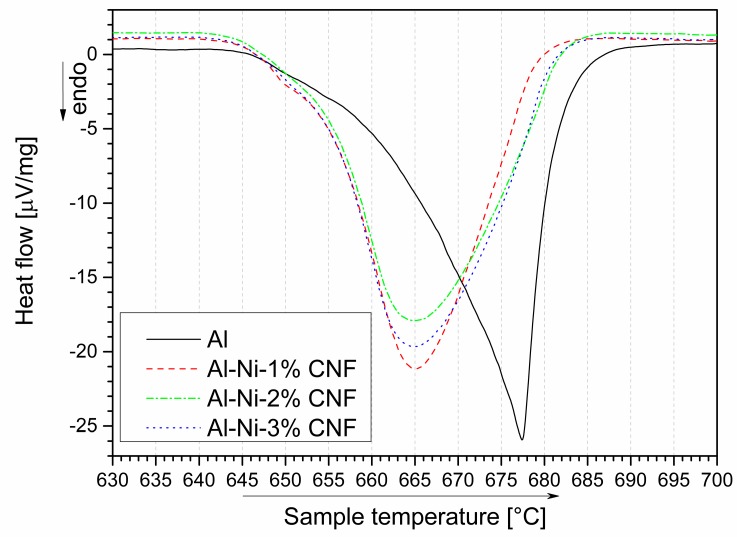
DSC curves of aluminum with various CNF content during heating near Al melting point.

**Figure 5 nanomaterials-09-00550-f005:**
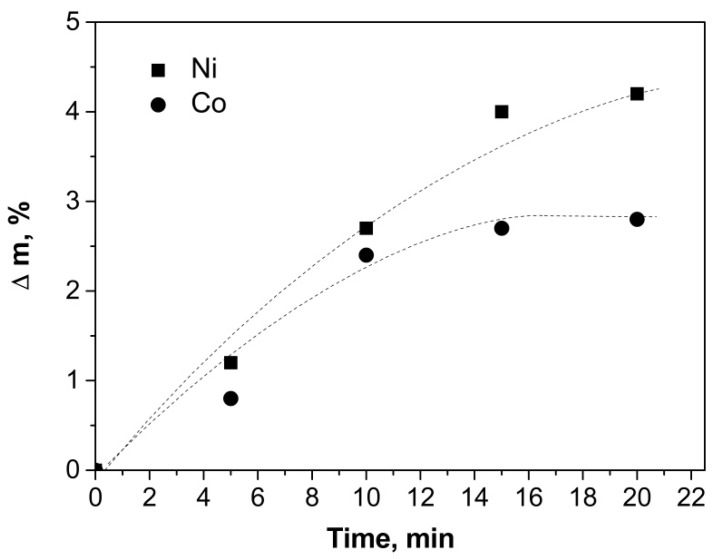
Dependence of the specimen mass variation on the synthesis time at 550 °C with Ni and Co catalysts.

**Figure 6 nanomaterials-09-00550-f006:**
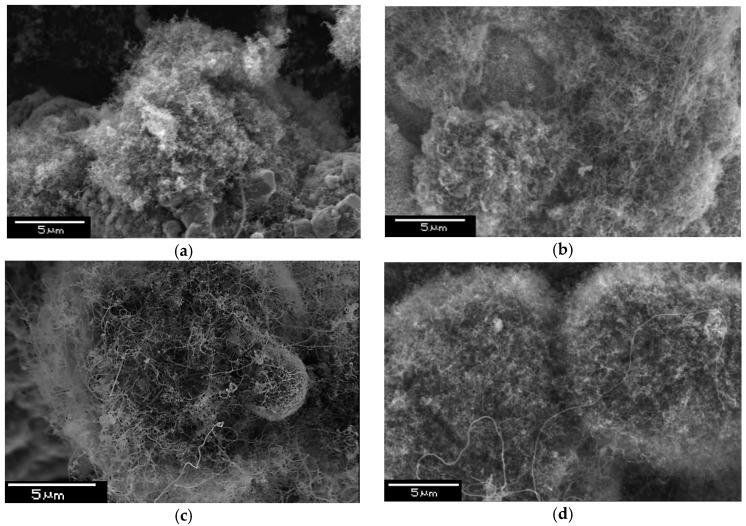
Microstructure of composites synthesized with Ni catalyst (**a**) and Co catalyst (**b**) (synthesis time 10 min) and with Ni catalyst (**c**) and Co catalyst (**d**) (synthesis time 20 min).

**Figure 7 nanomaterials-09-00550-f007:**
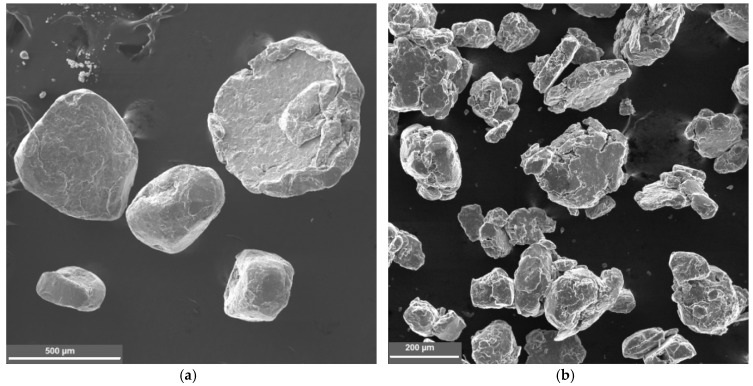
Microstructure of composites after processing in planetary mill (**a**) and carbon distribution in composite particles (**b**).

**Figure 8 nanomaterials-09-00550-f008:**
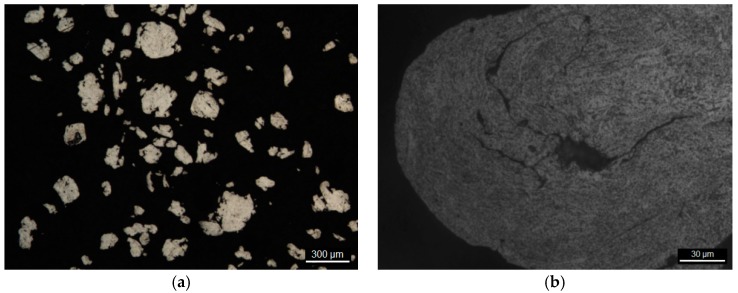
Structural characteristics of the composite particle microstructure. Microscope magnification ×50 (**a**) and ×500 (**b**).

**Figure 9 nanomaterials-09-00550-f009:**
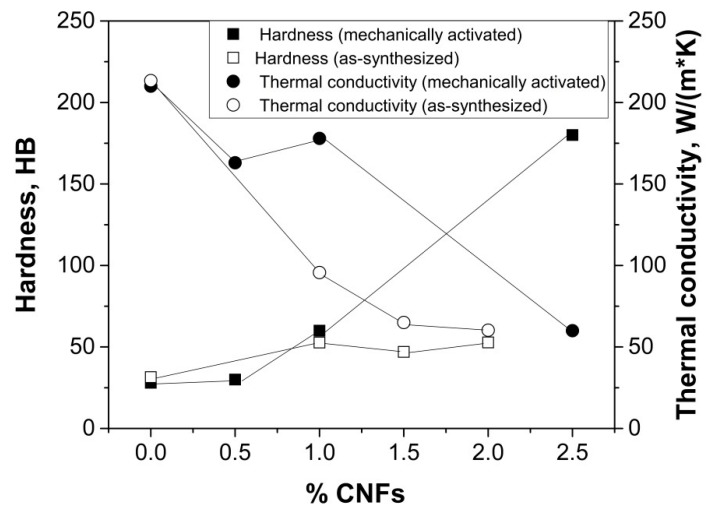
Hardness and thermal conductivity versus carbon content.

**Figure 10 nanomaterials-09-00550-f010:**
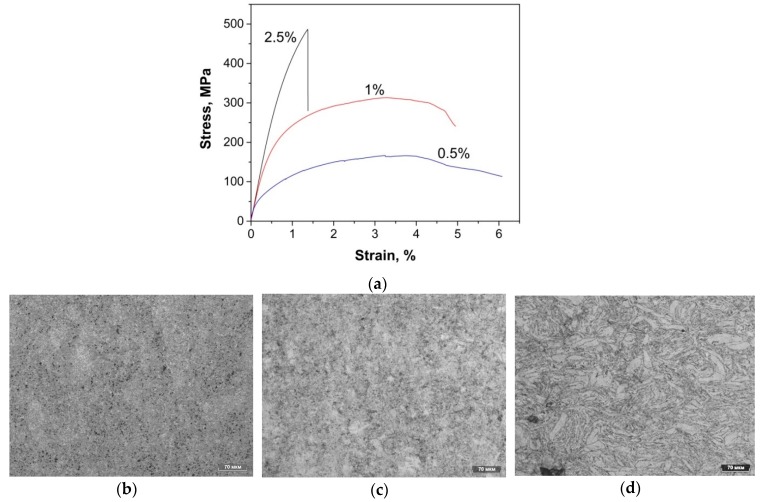
The results of three point flex test of Al-CNFs composites (**a**) and optical micrographs of the polished cross sections of (**b**) 2.5 wt.%, (**c**) 1 wt.%, and (**d**) 0.5 wt.% Al–CNFs composites.

**Table 1 nanomaterials-09-00550-t001:** Specific surface and porosity of aluminum/carbon nanofibers (CNFs) samples.

Specimen	Specific Surface Area, m^2^/g (BET, Kr, 77 K)	Specific Surface Area, m^2^/g (BET, N_2_, 77 K)	Specific Porosity, cm^3^/g (Gurvich, N_2_, 77 K)
Al-Ni	0.125	-	-
Al-Ni-1%CNFs	3.71	3.58	0.014
Al-Ni-1%CNFs+flux	0.42	0.42	0.003
Al-Ni-2%CNFs	4.70	4.26	0.023
Al-Ni-3%CNFs	5.04	3.85	0.013

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
