# Peer review of "Conditions for Production of Composite Material Based on Aluminum and Carbon Nanofibers and Its Physic-Mechanical Properties"

_nanomaterials, 2019, doi:10.3390/nano9040550_

Round 1
Reviewer 1 Report
Review for Tolochko, O.V. et al., “Conditions for Production of Composite Material based on Aluminum and Carbon Nanofibers (CNFs) and Its Physic-mechanical Properties” Title: Please do not introduce abbreviations (CNF) in the title Keywords: Please use more specific keywords 1. Introduction: I doubt that the introduction is specific enough for the target group of your article. I suggest rewriting it and focusing on the composite properties which were investigated. I do not agree with most part of the introduction and list some points of critics as suggestions: - I personally do not know of attempts to introduce CNF-reinforced aluminum to either aerospace or automotive industry. - The unique properties of CNF may be true but at least for thermal conductivity, none if really improves the composite, which is shown in your paper. - uniform distribution is pointed out as one main problem for CNF reinforced aluminum. What about poor interface quality, porous structure and voids? Not having a fully dense material will influence its ductility much more than a non-uniform distribution of CNFs. Typos: L35: “is the development of” L48: “A significant problem here is” 2. Materials and Methods L73: Please add the approximate volume of the reactor. Otherwise, the flow rate given is meaningless. L80: “with a certain amount” L82: “were compacted by cold pressing” Powder Compaction: Did all specimens undergo this compaction? How many specimens were prepared for each group? L89: “by equation (1)” L90: where cp ([cp] = J(gK)) L91: how were the accuracies given? From datasheets of manufacturer L99: “data obtained using” 3. Results and Discussion L113: CNFs assist surface and porosity increase. → Probably “assist” has a too positive connotation, since porosity is definitely not a preferable property of a composite. Table 1: how many specimens were analysed for the mean values? Is Al-Ni totally free of pores or was it not measured? How come, 2% and 3% specimens were not tested with added flux? How is the decrease in porosity explained, when stepping up from 2% to 3% CNF content? Figure 2,3,4 → please use white/transparent background. Distinguishing the graphs by using different strokes/line-types is highly advised for b/w printout. Figure 6 and discussion: how were the differences in carbon content and length of the carbon structure quantified? How many samples were analysed? Are they representative? L167: please use “µm” instead of “microns” L172: how was the microhardness of the 50-200µm particles tested? L177: “was activated by hot pressing” L194: “illustrates the bending test results” Figure 9: While I can think of many applications for aluminum with increased thermal conductivity, I can hardly think of any application where defining the thermal conductivity at a certain lower range would be of value. Figure 10b why is the magnification different here? For the sake of better comparability, please keep the magnification constant. Was the porosity of the bending samples tested? This will most likely elucidate further on the poor ductility with the high CNF-content. “4. Conclusion” L222: “provides good nanofiber distribution” L226f should be: “strength, ductility and thermal conductivity … can be varied by introducing CNF to pure aluminum in different concentrations” The conclusion is somewhat contradictory to the introduction. No increase in thermal conductivity is reached. Even though the distribution is claimed to be uniform, ductility is decreasing, when strength is increasing. The strong point of the paper is actually the introduction of the flux, however, too few comparisons of with and without flux were made. There is room for improvement.
Author Response
Responses to reviewers ' comments
Marking (Corrected ) means change in text ( in yellow)
Reviewer 1 | |
Please do not introduce abbreviations (CNF) in the title | Corrected |
Keywords: Please use more specific keywords | Corrected |
I doubt that the introduction is specific enough for the target group of your article. I suggest rewriting it and focusing on the composite properties which were investigated. I do not agree with most part of the introduction and list some points of critics as suggestions: - I personally do not know of attempts to introduce CNF-reinforced aluminum to either aerospace or automotive industry. - The unique properties of CNF may be true but at least for thermal conductivity, none if really improves the composite, which is shown in your paper. - uniform distribution is pointed out as one main problem for CNF reinforced aluminum. What about poor interface quality, porous structure and voids? Not having a fully dense material will influence its ductility much more than a non-uniform distribution of CNFs. | Corrected |
Typos: L35: “is the development of” L48: “A significant problem here is” | Corrected |
2. Materials and Methods L73: Please add the approximate volume of the reactor. Otherwise, the flow rate given is meaningless. | Corrected |
L80: “with a certain amount” L82: “were compacted by cold pressing” | Corrected |
Powder Compaction: Did all specimens undergo this compaction? How many specimens were prepared for each group? | Added |
L89: “by equation (1)” L90: where cp ([cp] = J(gK)) | Corrected |
L91: how were the accuracies given? From datasheets of manufacturer | The accuracies were given from datasheets of manufacturer and was checked by measuring reference samples. |
L99: “data obtained using” | Corrected |
3. Results and Discussion L113: CNFs assist surface and porosity increase. → Probably “assist” has a too positive connotation, since porosity is definitely not a preferable property of a composite. | Corrected |
Table 1: how many specimens were analysed for the mean values? | Corrected Three samples were measured to obtain the average values of the measured values. |
Is Al-Ni totally free of pores or was it not measured? | Average particle size of Al-Ni sample was fairly high, around 50 μm. So it's porosity accessible for gas adsorption turned out to be negligable.
|
How come, 2% and 3% specimens were not tested with added flux? | After receiving the data on the sample with 1% it became clear that the addition of flux does not lead to the desired result ( increase in the specific surface area), it was decided not to study other samples. |
How is the decrease in porosity explained, when stepping up from 2% to 3% CNF content? | We consider results of nitrogen adsorption to be rather rough. While measurements of N2 adsorption at 77 K are widely used their main disadvantage is low sensitivity. For presice determination of SSA krypton adsorption at 77 K was used. Owing to high saturated vapour pressure of Kr such measurements are much more accurate but at the same time are limited to relative pressure 0.30-0.35 while Gurvich pore volume is determined in 0.95-0.99 relative pressure range. Different precision of nitrogen and krypton can be clearly seen from specific surface area values in Table 1. Nevertheless, we decided to leave N2 result in order to give researchers a possibility to compare results with other materials since N2 (unlike Kr) is virtually a standart adsorptive.
|
Figure 2,3,4 → please use white/transparent background. Distinguishing the graphs by using different strokes/line-types is highly advised for b/w printout. | Corrected |
Figure 6 and discussion: how were the differences in carbon content and length of the carbon structure quantified? How many samples were analysed? Are they representative? | |
The carbon content was determined initially by TGA and for all samples just by weighing before and after synthesis. The length of carbon nanostructures was determined visually by SEM images. Added | |
L167: please use “µm” instead of “microns” | Corrected |
L172: how was the microhardness of the 50-200µm particles tested? | The load during indentation was 1 g and the diagonal of the impression was less 10 μm. Microhardness was determined on cross sections of particles. |
L177: “was activated by hot pressing” | Corrected “Mechanical activated powder material after with flux was compacted by the hot pressing.” |
L194: “illustrates the bending test results” | Corrected |
Figure 9: While I can think of many applications for aluminum with increased thermal conductivity, I can hardly think of any application where defining the thermal conductivity at a certain lower range would be of value. | Added |
Figure 10b why is the magnification different here? For the sake of better comparability, please keep the magnification constant. | Corrected |
Was the porosity of the bending samples tested? This will most likely elucidate further on the poor ductility with the high CNF-content. | Added: The relative density of the samples, determined by the method of hydrostatic weighing, was not less than 97%. |
“4. Conclusion” L222: “provides good nanofiber distribution” L226f should be: “strength, ductility and thermal conductivity … can be varied by introducing CNF to pure aluminum in different concentrations” | Corrected |
The conclusion is somewhat contradictory to the introduction. No increase in thermal conductivity is reached. Even though the distribution is claimed to be uniform, ductility is decreasing, when strength is increasing. The strong point of the paper is actually the introduction of the flux, however, too few comparisons of with and without flux were made. There is room for improvement. | Corrected |

Reviewer 2 Report
The authors investigated the PM production of aluminum reinforced with carbon nanofibres. Quality of presentation of methods a well as is high.
The author write Figure 6a - l. 171; is it not Figure 7a ?
Similar: l. 182; the authors write [27] - and not [26] ?
Did the authors consider (calculate?) the density of newly created dislocations after cooling from an elevated temperature to room temperature?
Author Response
Responses to reviewers ' comments
Marking (Corrected ) means change in text ( in yellow)
Reviewer 2 | |
The author write Figure 6a - l. 171; is it not Figure 7a ? | Corrected |
l. 182; the authors write [27] - and not [26] | We lost [26] in L44 |
Did the authors consider (calculate?) the density of newly created dislocations after cooling from an elevated temperature to room temperature? | Thank you very much for suggestion we just suppose to check that, however In recent article, this issue is not addressed. |

Round 2
Reviewer 1 Report
N/A